# Long-Term Follow-Up of Empirical Slow Pathway Ablation in Pediatric and Adult Patients with Suspected AV Nodal Reentrant Tachycardia

**DOI:** 10.3390/jcm12206532

**Published:** 2023-10-15

**Authors:** Marta Telishevska, Sarah Lengauer, Tilko Reents, Verena Kantenwein, Miruna Popa, Fabian Bahlke, Florian Englert, Nico Erhard, Isabel Deisenhofer, Gabriele Hessling

**Affiliations:** Department of Electrophysiology, German Heart Center Munich, Technical University of Munich, Lazarettstr. 36, 80636 Munich, Germany; lengauer@dhm.mhn.de (S.L.); reents@dhm.mhn.de (T.R.); kantenwein@dhm.mhn.de (V.K.); popa@dhm.mhn.de (M.P.); bahlke@dhm.mhn.de (F.B.); englert@dhm.mhn.de (F.E.); erhard@dhm.mhn.de (N.E.); deisenhofer@dhm.mhn.de (I.D.); hessling@dhm.mhn.de (G.H.)

**Keywords:** atrioventricular nodal reentry tachycardia, dual atrioventricular nodal physiology, empirical slow pathway ablation, recurrence

## Abstract

Background: The aim of this study was to assess long-term efficacy and safety of empirical slow pathway (ESP) ablation in pediatric and adult patients with a special interest in patients without dual AV nodal physiology (DAVNP). Methods: A retrospective single-center review of patients who underwent ESP ablation between December 2014 and September 2022 was performed. Follow-up included telephone communication, letter questionnaire and outpatient presentation. Recurrence was based on typical symptoms. Results: 115 patients aged 6–81 years (median age 36.3 years, 59.1% female; 26 pts < 18 years) were included. A typical history was present in all patients (100%), an ECG documentation of narrow complex tachycardia in 97 patients (84%). Patients were divided into three groups: Group 1 without DAVNP (*n* = 23), Group 2 with AH jump (*n* = 30) and Group 3 with AH jump and at least one AV nodal echo beat (*n* = 62). No permanent AV block was observed. During a median follow-up of 23.6 ± 22.7 months, symptom recurrence occurred in 7/115 patients (6.1%) with no significant difference between the groups (*p* = 0.73, log-rank test). Symptom recurrence occurred significantly more often in patients without (5/18 patients; 27%) as compared to patients with ECG documentation (2/97 patients; 2.1%; *p* = 0.025). No correlation between age and success rate was found (*p* > 0.1). Conclusions: ESP ablation is effective and safe in patients with non-inducible AVNRT. Overall, recurrence of symptoms during long-term follow-up is low, even if no DAVNP is present. Tachycardia documentation before the EP study leads to a significantly lower recurrence rate following ESP ablation.

## 1. Introduction

Atrioventricular nodal reentry tachycardia (AVNRT) is the most common paroxysmal supraventricular tachycardia (SVT) in adolescents and adults. Catheter ablation is the treatment of choice for symptomatic AVNRT patients. However, the endpoint of the procedure is still not unequivocally established [1]. Catheter ablation for SVT in general, and AVNRT in particular, is the current treatment of choice for symptomatic patients because it substantially improves quality of life [2]. However, in some patients with typical symptoms, no SVT is inducible during the EP study and the final diagnosis of the tachycardia mechanism remains undetermined. In these patients, the ACC/AHA/ESC Guidelines [3] suggest an empirical slow pathway (ESP) ablation under the precondition of (1) existence of a Holter or surface electrocardiographic (ECG) documentation of a tachycardia compatible with AVNRT and (2) documentation of dual atrioventricular nodal pathway physiology (DAVNP) in the form of an AV nodal echo beat and/or an AH jump during programmed atrial stimulation during the EP study.

Studies in adults and some smaller studies in pediatric patients have reported on the long-term outcome of ESP modification under these preconditions [4,5,6,7,8,9,10]. Data are lacking about ESP modification in patients with suspected AVNRT without DAVNP during the EP study.

The main threshold to perform ESP ablation in suspected AVNRT is due to the low but not zero risk of ablation-related complete AV block. The clinical relevance of this dilemma is highlighted by a recent survey [7]. In the absence of a characteristic ECG documentation, 44% of the interviewed electrophysiologists stated that it would require a minimum of two echo beats as a threshold to perform ESP, while 19% did not even consider two echo beats as a sufficient indication for slow pathway modification [7]. In another study investigating procedural success and clinical long-term outcome after ESP ablation performed with a minimum of two AV nodal echo beats, the threshold to ablate was lower when a characteristic ECG documentation was present [4].

The aim of this study was to evaluate the long-term outcome after ESP ablation in a large cohort of pediatric and adult patients with suspected, but non-inducible, AVNRT with and without DAVNP in the EP study.

## 2. Methods

### 2.1. Study Design

This study was a retrospective analysis of 115 consecutive patients (26 patients < 18 years) who underwent ESP ablation at our institution from December 2014 to September 2022. Baseline characteristics are shown in Table 1. Patients with accessory pathways or with inducible AVNRT during the EP study or during RF delivery at the slow pathway region were excluded.

All patients fulfilled the following criteria: (1) typical history (typical on—off palpitations) and/or ECG documentation (narrow complex tachycardia with regular cycle length and no discernible P-wave; (2) non-inducibility of AVNRT or other SVT by programmed atrial stimulation with and without pharmacological stimulation; (3) completion of a telephone questionnaire for long-term follow-up.

Patients were divided into 3 groups: Group 1 included patients without AH jump and without AV nodal echo beat(s), Group 2 patients with AH jump without AV nodal echo beat and Group 3 patients with AH jump and 1–2 AV nodal echo beat(s) during the EP study.

The Institutional Ethical Review Board approved the study.

#### 2.1.1. Electrophysiological Study

Electrophysiological studies were performed after written informed consent. All antiarrhythmic drugs had been discontinued at least three half-lives prior to the procedure. The procedure was performed under general anesthesia in pediatric patients < 45 kg (*n* = 3), analgesia/sedation with propofol and fentanyl (*n* = 108) or analgesia only with fentanyl (*n* = 4).

The procedures were performed by four operators with more than 10 years of slow pathway ablation experience utilizing electroanatomical mapping (EAM) systems.

Diagnostic catheters were inserted via transfemoral access and positioned in the coronary sinus, His bundle area, and right ventricle. After placement of catheters, a bolus of 5000 U heparin (pediatric dose 100 U/kg) was administered intravenously. The presence of an accessory pathway was excluded, as described previously [9].

AVNRT induction was attempted using programmed atrial stimulation with 2 basic cycle lengths (600/400 ms or 500/400 ms) followed by 1–3 extrastimuli and additionally by atrial burst stimulation up to a minimum CL of 200 msec. If no tachycardia was induced, orciprenaline (since 2020 isoproterenol) was administered intravenously aiming at a heart rate increase of at least 20% and the stimulation protocol repeated.

Dual atrioventricular nodal physiology (DAVNP) was defined as the presence of any of the following: (1) jump phenomenon as a prolongation of the AH-interval by more than 50 msec after a 10 msec decrease in the coupling interval during programmed extrastimulation; (2) typical slow–fast atrioventricular nodal echo beat; and (3) sustained slow pathway conduction seen during rapid atrial pacing, as described previously [8].

#### 2.1.2. Ablation

A right atrial map was performed using a 3D mapping system in all patients. The expected location of the rightward inferior slow pathway extension was mapped in sinus rhythm at the triangle of Koch, between the posteroseptal tricuspid valve annulus and the ostium of the coronary sinus. Mapping and ablation were performed using a non-irrigated 4-mm tip catheter (Marinr^®^ 7F, Medtronic, Minneapolis, MN, USA; 60°/30W or Navistar^®^ 7F, Biosense, Webster, TX, USA). The catheter was delivered through a standard 8F venous sheath. A long introducer sheath (SR0 ™Fast-Cath™ 8.F, Abbott or SL0 ™Swartz™ 8.5F, Abbott, TX, USA) was used at the discretion of the operator if no stable mapping position was achieved. Non-irrigated RF ablation was targeted at atrioventricular groove sites with a dominant ventricular electrogram and ideally with a “bump-and-spike” atrial electrogram signifying a near-field slow pathway potential. Test radiofrequency applications at target sites were continued for up to 10 s in search junctional rhythm as feedback for effective SP modification. If junctional rhythm at a temperature ≥ 48 °C was achieved, applications were continued for at least 60 s. Applications were terminated if either rapid junctional acceleration (cycle length < 400 ms) or PR interval prolongation was noticed. System impedance measured over the ablation catheter was monitored for evidence of effective tissue heating (10–15 Ω decrease), inadvertent catheter slippage into the coronary sinus or coagulum formation.

#### 2.1.3. Empirical Slow Pathway Ablation

The primary ablation objective was slow junctional rhythm at the target area and a catheter-tip temperature of ≥48 °C. If junctional rhythm with good temperature was noted, RF applications of at least of 60 s were delivered. If no junctional rhythm was observed at various sites, applications were applied corresponding to electrical signals and anatomy. In Group 2 and 3, ablation aimed at eliminating the AH jump and echo beats. However, if lesions were considered effective, as described above, a residual AH jump with one echo beat was tolerated. The atrial stimulation protocol was repeated 20 min following the last ablation.

#### 2.1.4. Post-Ablation Care and Follow-up

In all patients, a transthoracic echocardiogram was performed at the end of the procedure and before discharge. A 12-lead ECG was obtained at the end of the procedure, on the evening of the ablation day and on the day following ablation. A 24-h Holter ECG was obtained before discharge.

In all adult patients, a purse-string suture was applied to venous puncture sites. Venous sheaths were removed directly after purse-string suture, and a groin compression bandage was applied for 6 h. In pediatric patients, venous puncture site compression was performed in the EP lab and a groin compression bandage was implemented for 6 h. All patients received heparin i.v. until removal of bandages. On the day after the ablation procedure, all punctured sites were carefully examined. In the case of abnormalities, an ultrasound evaluation of the femoral vessels was performed.

Follow-up was conducted in our outpatient clinics and by telephone communication with the patient. Clinical outcome was assessed by absence/recurrence of clinical symptoms and/or ECG documentation. Routine ambulatory monitoring was not performed in asymptomatic patients. Time to recurrence was defined as time between procedure and documented tachycardia or recurrence of typical symptoms. Study endpoints were complete elimination of clinical symptoms without documentation of AVNRT recurrence.

## 3. Statistical Analysis

Mean values were calculated as arithmetic averages and represented as the mean ± SD. Comparisons between groups were made by an unpaired *t* test or one-way analysis of variance for normally distributed variables. A value of *p* < 0.05 was considered statistically significant. SPSS 23 (SPSS Inc., Chicago, IL, USA) was used for all calculations. For patients with follow-up after initial ablation, freedom from AVNRT recurrence was estimated using the Kaplan–Meier method.

## 4. Results

### 4.1. Baseline and Electrophysiological Characteristics

Clinical and electrophysiological characteristics of the patients are shown in Table 1. All patients described typical on—off palpitations and 84.3% of patients (97/115) had ECG documented regular narrow complex tachycardia. Six patients had received a prior EP study without catheter ablation. Median age was 36.33 ± 18.91 years (range 6–81) with 68/115 female patients (59.1%). In four patients, congenital heart disease was present, including cc-TGA with VSD after pulmonary artery banding and tricuspid valve replacement (*n* = 1), bicuspid aortic valve with aortic insufficiency (*n* = 1), aortic stenosis after aortic valve replacement (*n* = 1) and mitral valve disease with mitral insufficiency (*n* = 1). During the EP study, 23/115 pts (20%) showed absence of DAVNP (Group 1), and 30/115 pts (26%) had an AH jump without AV nodal echo beat (Group 2). In 62/115 (54%) patients, DAVNP with AH jump and at least one AV nodal echo beat (7/62 two echo beats) was present. Atrial fibrillation occurred spontaneously or after programmed stimulation in 7/115 (6.6%) patients. No ECG documentation of narrow complex tachycardia was present in 7/23 pts (30.4%) from Group 1, in 5/30 pts (16.7%) from Group 2 and 6/62 pts (9.6%) from Group 3.

### 4.2. Radiofrequency Ablation

A 3D mapping system was utilized in all patients (n = 115) including the Ensite Velocity/Precision/X™ System (Abbott, TX, USA) in 99 patients (86.1%) or the Carto^®^3 System (Biosense Webster, TX, USA) in 16 patients (13.9%). In five patients (4.8%), the procedure was performed without fluoroscopy. Junctional rhythm during ablation was noted in 104/115 (90.4%) patients. In Group 1, in 18/23 (78%) patients junctional rhythm was noted during ablation, in Group 2 in 27/30 (90%) patients and in Group 3 in 59/62 (95%) patients (*p* > 0.1). In Group 2, no residual post-ablation AH jump was documented in any patient. In Group 3, a residual AH jump (*n* = 5 pts) or a residual AH jump with one echo beat (*n* = 6 pts) was present.

### 4.3. Procedure/Ablation Times and Fluoroscopy Time/Dose

Mean procedure time was 81.5 ± 26.3 min (38–170 min) with a mean fluoroscopy time of 3.4 ± 2.8 min (range 0.5–12.5 min) and a mean fluoroscopy dose of 108.7 ± 162.7 cGycm^2^ (range 3.78–515.9). Mean RF-time was 4.2 min. Procedure time, fluoroscopy time/dose and RF-time were not significantly different between the groups. Procedural and ablation data are shown in Table 2.

### 4.4. Safety

Overall mortality was 0%. No intermittent or permanent complete AV block occurred. In Group 3, one major complication was noted in a 69-year-old woman with multiple chronic comorbidities as COPD GOLD IV, postrenal kidney failure, pulmonary embolism and a history of multiple embolic cerebral infarctions. Pericardial effusion was detected 4 h after ablation, probably due to coronary sinus perforation. After successful pericardiocentesis the patient was hemodynamically stable. The pericardial sheath was removed the next day and the patient was discharged 5 days after the ablation in stable condition. In all patients, transthoracic echocardiography at discharge documented normal ventricular function without signs of dyskinesia, thrombi or pericardial effusion. Minor complications were observed in seven patients (6.1%). These included transient first- or second-degree atrioventricular block (*n* = 4) or right bundle branch block (*n* = 3), completely resolving by the end of the electrophysiological study. No vascular complications were observed.

### 4.5. Follow-up

Over a mean observation time of 23.67 ± 22.7 months (3–92 months), seven of 115 patients (6.1%) had recurrence of clinical symptoms. Symptom recurrence occurred in 2/23 patients (8.7%) in Group 1, 2/30 patients (6.7%) in Group 2 and 3/62 patients (4.8%) in Group 3 (Table 3).

Kaplan–Meier analysis shows freedom from AVNRT recurrence without significant difference between the three groups (*p* = 0.73, log-rank test; Figure 1).

No junctional rhythm at ablation had been noted in 4/7 pts (2/2 from Group 1, 2/2 in Group 2 and 0/3 in Group 3) with symptom recurrence. Of the seven patients with symptom recurrence, five patients did not want a second ablation due to rarity of symptoms. Two patients underwent a second SP ablation and interestingly demonstrated AVNRT-induction during the redo procedure (one patient from Group 1 and one patient from Group 2, all without junctional rhythm during the first ablation).

Recurrence rate was significantly higher in patients without prior ECG documentation (5/18 patients; 27%) than in patients with pre-ablation ECG documentation (2/97 patients; 2.1%; *p* = 0.025). In Group 1, 2/2 pts with recurrence from Group 1 were without ECG documentation, 1/2 pts from Group 2, and 2/3 pts from Group 3. This is shown in Table 4.

Pearson’s correlation coefficient shows no statistical relationship between younger age and longer RF time (r = 0.03) and no correlation between age and success rate of ESP ablation (*p* > 0.1) was found. All 26 pediatric patients and the six patients with prior EP study remained free from symptom recurrence after ESP ablation.

Table 5 shows clinical details of patients reporting symptom recurrence after ESP ablation. No patient required antiarrhythmic drug treatment during follow-up.

## 5. Discussion

This large study with 115 patients evaluates empirical slow pathway ablation in symptomatic pediatric and adult patients without inducible AVNRT at the EP study. It is the first study that includes patients with a clinical history consistent with AVNRT but without evidence of dual AV nodal physiology at the EP study. Empirical slow pathway ablation was performed by experienced operators at a high-volume center.

The main finding of the study is that ESP ablation is effective for patients with non-inducible AVNRT and DAVNP, but also for patients without DAVNP in the EP study. The approach is safe and symptom recurrence rate is low, especially in patients with a clinical history consistent with AVNRT and ECG documentation of a regular narrow complex tachycardia before the EP study.

Patients with clinical history consistent with AVNRT but without inducible AVNRT in the EP study still present a problem, as the operator has to decide whether to proceed with ablation. Sluggish antegrade slow-pathway conduction or intermittent suppression of DAVNP due to sedation or other transient factors during the EP study may mask DAVNP. AVNRT inducibility is sometimes complex and requires a perfect balance between slow and fast pathway conduction. However, ESP ablation in non-inducible patients is the only option for a potential cure even in the absence of a clear pre-EP study of short RP tachycardia or DAVNP in an EP study [11].

In a retrospective study by Gerguri et al. [5], 63 patients (19%) with no pre-procedural ECG documentation but DAVNP with a maximum of one echo beat in the EP study were assigned to a “pure” empirical SP group. The other 271 patients (81%) with ECG documentation were assigned to the standard SP group. A higher incidence of other tachycardias and a higher persistence of subjective symptoms were found after pure empirical SP ablation. However, clinical symptoms improved in 60% of patients. The study by He et al. [12] had reported that SP ablation can be performed safely and effectively in non-inducible and suspected AVNRT patients with good long-term results. Also in this study, no tachycardia documentation before ablation was predictive for recurrence. In our study, only 15.6% of patients had no ECG documentation of narrow complex tachycardia. Symptom recurrence was significantly higher in those patients. This be due to the fact that other tachycardia mechanisms (especially focal atrial tachycardia) play a role and ablation did not meet the “real” target.

Our primary ablation objective and endpoint for ablation was sustained slow junctional rhythm with a catheter-tip temperature of ≥48 °C. This sustained junctional rhythm as procedural “endpoint” was noted in 90.4% of patients. Shurrab et al. also reported about 90.7% junctional rhythm performing SP ablation [11]. We suggest that slow junctional rhythm could serve as a sufficient endpoint in these cases. Some studies decided DAVNP was a recognized reliable marker of success [13,14], but their results did not show a significant difference to long-term success between junctional rhythm alone versus junctional rhythm plus abolishment of DAVNP as ablation endpoints [14]. Operators do not necessarily need the greater effort and therefore have the greater risk in trying to achieve DAVNP abolishment. The study by Duman et al. showed a non-inducible AVNRT may be accompanied by the absence of some features that represent DAVNP, such as jump, and/or echo beats. The absence of any of these dual AV node features, established before ablation, did not have any effect on outcome [9].

The study by Duman et al. found a procedural success rate of 97.4% and a recurrence rate of 12.6% [9]. In the recent study by O’Leary et al. [15] with 512 pediatric patients with SP ablation for inducible AVNRT, the recurrence rate was 5%. In our study, symptom recurrence after ESP ablation was observed in only 6.1% of patients, which is comparable to the study of Duman. Two patients underwent a second SP ablation and demonstrated AVNRT-induction during a redo of SP ablation; in 5/7 patients, there was a decrease in tachycardia episodes following ESP ablation, and no second ablation was performed [9,16,17].

The main aspects of ESP ablation are safety and efficacy. Although not used in our study, intracardiac echocardiography (ICE) is a technique to avoid or minimize fluoroscopy and increase safety. Potential advantages over 3D EAM are the real-time visualization of true endovascular borders (including duplex-mediated blood flow direction), endocardial structures and diagnostic or ablation catheters, as well as immediate recognition of potential complications (such as pericardial effusion or thrombus formation) [18]. The study by Luani et al. reported that zero-fluoroscopy ICE-guidance shows comparable efficacy and safety when compared to traditional fluoroscopic navigation during cryothermal ablation of the slow pathway in AVNRT patients [18].

The study by Bocz et al. with 80 patients who underwent SP ablation for AVNRT showed no recurrences during follow-up. This study compared ICE guidance and EAM system guidance. Utilizing ICE for anatomical SP ablation showed notable advantages over EAMS-guided procedures [19]. Another study by Kupo et al. demonstrated that ICE-guidance during SP ablation significantly reduces mapping and ablation time, radiation exposure, and RF delivery in comparison to fluoroscopy-only procedures. Moreover, early switching to ICE-guided ablation seems to be an optimal choice in challenging cases [20]. In summary, ICE might present a good alternative to conventional EAM in SP ablation.

In the study by Duman et al. [9] with 79 pediatric patients, the higher prolongation in Wenckebach cycle length and lower mean age were predictors for AVNRT recurrence. In the recent adult study by He et al. [12], a younger age (adolescent age/14–18 years) was related to a higher long-term recurrence rate. We did not find a correlation between age and longer RF time or long-term efficacy of ESP ablation. All pediatric patients in our study showed freedom from symptom recurrence after ESP ablation.

A zero/minimal fluoroscopy approach should be pursued especially in pediatric cases. In our study, in five patients the procedure was performed without fluoroscopy using the Ensite/NavX system. In all the other cases, we aimed for minimal fluoroscopy. In the study by Smith and Clark, zero fluoroscopy was possible in 80% of the procedures, and the use of EAM resulted in a significant decrease in fluoroscopy time [21]. Walsh et al. achieved a complete zero-fluoroscopic approach in 94% (in 47 of 50 pts.) using EAM [22]. The mean fluoroscopy time in our study was higher (3.4 ± 2.8 min) in comparison to the study by Swissa et al. (0.83 ± 1.04 min) using a fluoroscopy-limited approach with a 3D EAM system [23]. The importance of reducing radiation exposure and achieving minimal/zero fluoroscopy procedures is highlighted in the study by Gaita F et. al. [24]. This review summarizes the state of the art of feasibility and safety of the near-to-zero approach for electrophysiological procedures. A meta-analysis by Debreceni et al. reported that a zero/minimal fluoroscopy approach using 3D EAM systems for catheter ablation of SVTs is feasible without compromising acute and long-term success or complication rates [25].

The six patients with previous EP studies with non-inducible AVNRT who underwent ESP ablation also showed freedom from symptom recurrence. In the absence of inducible SVT, some operators may hesitate to perform ESP ablation given the absence of a clear procedural endpoint and the low but not zero risk of complete AV block [26,27]. The difficult decision of whether to perform ESP ablation is a challenge in clinical practice and the perceived need for further evidence to guide this decision is high [7]. Our data suggest that it might be a valid option to extend the indication for ESP ablation beyond the current guidelines to those patients who exhibit DAVNP at the EP study in the absence of tachycardia ECG documentation and to those patients with a clinical history consistent with AVNRT and ECG documentation of tachycardia in the absence of DAVNP. In both cases, it might be advantageous to proceed to ESP ablation to avoid repeat EP studies including possible complications.

## 6. Conclusions

This study demonstrates that empirical slow pathway ablation in patients with a clinical history consistent with AVNRT with and without tachycardia ECG documentation is a safe procedure with a beneficial outcome for the great majority of the patients. For the first time, it is shown that even patients who do not present with dual-nodal AV physiology in the EP study benefit from the procedure, especially if ECG documentation of narrow complex tachycardia is available. This finding is independent of age; symptomatic children and adolescents without inducible tachycardia may also benefit from empirical slow pathway ablation. Our study might encourage electrophysiologists in high-expertise centers to perform ESP ablation in patients with a clinical history consistent with AVNRT and ECG documentation but without inducible AVNRT even if no DAVNP at the EP study is present.

## 7. Limitations

The main limitation is the retrospective character of the single-center cohort study with data collection over a longer time period which might include a possible variation in patient management over time. Intravenous sedation as performed at our institution may affect tachycardia inducibility. The use of ICE might further increase safety and efficacy but is currently not available at our center. The recurrence rate was assessed based on clinical parameters only. We cannot completely exclude a placebo effect after ESP ablation.

## Figures and Tables

**Figure 1 jcm-12-06532-f001:**
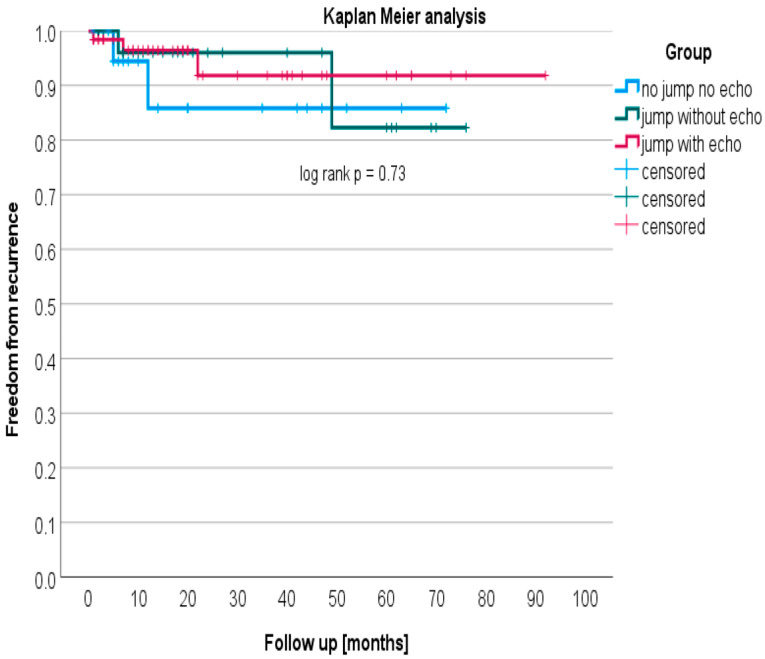
Kaplan–Meier analysis.

**Table 1 jcm-12-06532-t001:** Baseline Characteristics.

Variable	All (*n* = 115)	Group 1 *n* = 23 (20%)	Group 2 *n* = 30 (26%)	Group 3 *n* = 62 (54%)
Age [yrs], mean ± SD	36.3 ± 18.9	36.8 ± 19.9	34.1 ± 20.9	37.2 ± 17.7
Gender [% female]	68 (59.1)	12 (52.1)	16 (53.3)	40 (64.5)
Patients < 18 yrs (%)	26 (22.6)	4 (17.3)	9 (30)	13 (20.9)
ECG Documentation *n* (%)	97 (84.3)	16 (69.5)	25 (83.3)	56 (93.3)
Previous EP study without ablation	6 (5.2)	0	2 (6.6)	4 (6.4)
General anesthesia *n* (%)	3 (2.6)	2 (8.7)	0	1 (1.6)
Conscious sedation *n* (%)	108 (93.9)	21 (91.3)	27 (90.0)	60 (96.8)
Retrograde conduction *n* (%)	109 (94.8)	17 (73.9)	18 (60.0)	56 (90.3)
APERP [msec], mean ± SD	332 ± 62	330 ± 52	338 ± 70	334 ± 65
APERP msec after abl, mean ± SD	356 ± 83	357 ± 76	358 ± 93	351 ± 75
3D System *n* (%):	114 (99.1)			
NaVx	99 (86.1)	20 (86.9)	26 (86.7)	53 (85.4)
Carto 3	15 (13.0)	3 (13.0)	4 (13.3)	8 (12.9)
Atrial fibrillation *n* (%):	7 (46.6)	3 (13.0)	2 (6.7)	2 (3.2)

**Table 2 jcm-12-06532-t002:** Procedural Data.

Variable	All (*n* = 115)	Group 1 *n* = 23	Group 2 *n* = 30	Group 3 *n* = 62	*p* Value
Age [yrs], mean ± SD	36.3 ± 18.9	36.8 ± 19.9	34.1 ± 20.9	37.2 ± 17.7	>0.1
Procedure time [min], mean ± SD	81.5 ± 26.3	77.8 ± 28.9	82.9 ± 26.8	82.2 ± 25.4	>0.1
RF-time [min], mean ± SD	4.2 ± 2.8	3.2 ± 1.9	4.3 ± 2.4	4.6 ± 3.2	>0.1
Temperature [grad], mean ± SD	44.5 ± 3.2	44.2 ± 4.3	44.4 ± 2.3	44.6 ± 3.3	>0.1
RF-Energy [watt], mean ± SD	31.8 ± 26.1	26.6 ± 5.9	29.2 ± 5.6	35.3 ± 35.3	>0.1
Fluoro time [min], mean ± SD	3.4 ± 2.8	3.5 ± 2.9	2.7 ± 2.2	3.8 ± 2.9	>0.1
Fluoro dos. [cGycm^2^], mean ± SD	108.7 ± 162.7	119.5 ± 167.9	96.5 ± 170.3	110.8 ± 159.3	>0.1

**Table 3 jcm-12-06532-t003:** Symptom recurrence in regard to tachycardia ECG documentation.

Group	No ECG Documentation	Recurrence	Recurrence/No ECG Documentation
1	30.4% (7/23)	8.7% (2/23)	2/2
2	16.7% (5/30)	6.7% (2/30)	2/1
3	9.6% (6/62)	4.8% (3/62)	3/2

**Table 4 jcm-12-06532-t004:** Procedural characteristics and recurrence.

Variable	Recurrence *n* = 7	No Recurrence *n* = 108	*p* Value
Age [yrs], mean ± SD	33.4 ± 14.2	36.5 ± 19.2	0.60
No ECG Documentation *n* (%)	5 (71)	13 (12)	0.025
APERP after abl, mean ± SD	360.0 ± 105.0	357.5 ± 82.7	0.95
RF-time [min], mean ± SD	2.2 ± 1.3	4.4 ± 2.8	0.003
Temperature [grad], mean ± SD	46.2 ± 2.5	44.4 ± 3.3	0.14

**Table 5 jcm-12-06532-t005:** Clinical details of patients with symptom recurrence after ESP ablation.

	Group	Junctional Rhythm during Abl.	ECG Documentation before Abl.	Frequency of Symptoms before Ablation	ECG Documentation after Abl.	Frequency of Symptoms at FU	Second Ablation for AVNRT	Recurrency at Long-Term FU
Pat. 1	1	No	No	once a month	Yes	once a month	Yes, with AVNRT induction	No
Pat. 2	1	No	No	once a quarter	No	very rare	No	Yes
Pat. 3	2	No	No	twice a week	Yes	once a week	Yes, with AVNRT induction	No
Pat. 4	2	No	Yes	2–3 times/year	No	twice after abl.	No	Yes
Pat. 5	3	Yes	No	1–2 times/year	No	once after abl.	No	Yes
Pat. 6	3	Yes	No	once a month	No	once after abl.	No	Yes
Pat. 7	3	Yes	Yes	once a month	No	twice after abl.	No	Yes

## Data Availability

Not applicable.

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
