# Peer review of "Long-Term Follow-Up of Empirical Slow Pathway Ablation in Pediatric and Adult Patients with Suspected AV Nodal Reentrant Tachycardia"

_jcm, 2023, doi:10.3390/jcm12206532_

Round 1
Reviewer 1 Report
Overall, the manuscript presents an intriguing study on empirical slow pathway (ESP) ablation for patients with suspected atrioventricular nodal reentry tachycardia (AVNRT). The authors aim to assess the long-term outcomes of ESP ablation in a patient cohort, particularly those lacking inducible AVNRT during electrophysiological (EP) studies. Nevertheless, the manuscript exhibits several aspects that merit scientific criticism and warrant consideration for enhancement:
In the Introduction, the authors assert that "catheter ablation is regarded as first-line therapy for the treatment of symptomatic AVNRT, and slow-pathway ablation or modification is an accepted ablation endpoint." This statement is not entirely accurate. The strategy involves ablation of the slow pathway in cases of AVNRT. However, the standard endpoint for such procedures is tachycardia non-inducibility or a maximum of 1 echo beat.
How was the determination made that sustained slow junctional rhythm with a catheter-tip temperature of ≥ 48°C was a suitable ablation endpoint? Were there instances where this endpoint was achieved, yet clinical symptoms persisted?
How many operators conducted the ablations, and what was their level of experience in slow pathway ablations utilizing electroanatomical mapping systems? Was intracardiac ultrasound (ICE) employed in any cases?
The use of ICE has demonstrated enhanced procedural outcomes in slow pathway ablations compared to a strategy employing fluoroscopy alone (cryo: 10.1186/s12947-019-0162-2; RF: 10.1007/s10840-022-01126-y) and in comparison to electroanatomical mapping systems as well (doi.org/10.3390/jcm12175577). The manuscript should address the role of ICE in slow pathway ablations.
Zero-fluoroscopy procedures are gaining prominence in both pediatric and adult cases (DOI: 10.1093/eurheartj/ehw223; 10.1111/j.1540-8159.2007.00701.x; 10.1016/j.ijcard.2017.01.128; 10.1007/s00392-018-1220-8). Furthermore, a meta-analysis involving 9,074 patients has shown that the Zero/minimal fluoro approach for SVT treatment is a viable method that reduces fluoroscopy time, radiation exposure, and ablation duration, without compromising acute and long-term success rates or complication rates (10.3389/fcvm.2022.856145). Did the operators endeavor to achieve zero-fluoroscopy? This matter is especially critical in pediatric cases and necessitates discussion in the manuscript.
Could you elaborate on the potential placebo effect or psychological factors influencing symptom recurrence post-ESP ablation? Were data on patient-reported outcomes collected to shed light on subjective improvements?
In light of a subset of patients experiencing symptom recurrence post-ESP ablation, could you provide insights into refining patient selection criteria to optimize the success rate of the procedure? How can electrophysiologists discern patients most likely to benefit from this intervention?
The manuscript contains several grammatical and punctuation errors. Proofreading for language and style is indispensable to enhance the manuscript's overall quality.
Consider incorporating graphs, tables, or diagrams to elucidate key findings, study design, and statistical results. Visual representations can aid readers in comprehending complex information more readily.
In summary, the manuscript addresses a crucial topic but necessitates substantial improvement in terms of clarity, organization, critical analysis, and presentation. Addressing the aforementioned points would augment the manuscript's overall quality and impact.
Reviewer 2 Report
Telishevska and colleagues sought to present the outcomes of empiric slow pathway ablation in a population for whom the definition of “empiric” is expanded even further than previously. The study is retrospective and comes with a number of limitations. However, even when performed in a more empiric manner than previously performed, empiric slow pathway ablation appears to have promising results.
1. In group 1, with no AH jump, without AVN echo beats, no inducible SVT by programmed stimulation and no EKG documentation (23 patients), why was AVNRT suspected?
2. As this study does not have an untreated arm, the final statement is confusing. The statement says that a placebo effect cannot be completely excluded. What steps were taken to exclude a placebo effect?
Reviewer 3 Report
In the manuscript title “Long-term follow-up empirical slow pathway ablation in pediatric and adult patients with suspected AV nodal reentrant tachycardia”, the Authors performed a retrospective analysis assessing long-term safety and efficacy of radiofrequency ablation on pediatric and young adult patients with suspected, but non-inducible AVNRT with and without DAVNP at the EP study. Specifically, authors compare patients without AH jump and without AV nodal echo beats vs. patients with AH jump without AV nodal echo beat vs. patients with AH jump and 1-2 AV nodal echo beats.
The main result poses the attention on clinical advantage of slow pathway ablation also in patients without dual nodal AV physiology at the EP study.
The manuscript is well written and conclusions are consistent with results. Nevertheless, I have some comments:
1) was A-H and H-V intervals measured before and after ablation during EP study? Those intervals should be reported in table before and after ablation. Moreover, a significant or pathological AH/HV prolongation after catheter ablation should be described as adverse event procedure-related.
2) Authors often use the statement “patients with a typical AVNRT history”. Since Group 1 does not shows EP findings consistent with AVRT (but only ECG finding that might be related to AVNRT), I think it would me more appropriate say “patients with clinical history consistent with AVNRT”.
3) Conclusion: “This study demonstrates that empirical slow pathway ablation in patients with a typical AVNRT history with and without tachycardia ECG documentation is a safe procedure with a beneficial outcome for the great majority of the patients.” Please mitigate this statement introducing the consideration that it could be safe in high expertise centers. This message promotes the ablation in contrast with guidelines (for patients such as Group 1), increasing the risk of AV damage in less experienced operators.
4) I would appreciate a table describing symptoms before and after ablation in patients with symptoms recurrence at follow-up. Authors could be inspired by table 3 of the following article “Cryoablation of para-Hisian and mid-septal accessory pathways: long-term outcomes of a specific stepwise cryoablation protocol. Minerva Cardiol Angiol. 2023 Jun;71(3):333-341. doi: 10.23736/S2724-5683.22.06136-1”. Moreover, the mentioned article (despite focusing on septal accessory pathway cryoablation), focuses the attention on symptoms improvement as main outcome after ablation. I suggest authors to mention this article to corroborate the message that clinical outcome plays a more relevant role respect to a “pure” electrophysiological point of view, particularly in pediatric/young adult population.
5) The seven patients with recurrence had significant lower RF time (Table 4). Please, briefly discuss these findings. Is it possible that a lower RF time could be related to a safer approach due to absence of documented AVNRT during the EP study?
Round 2
Reviewer 1 Report
The authors have not adequately responded to my critical comments, and the requested revisions have not been made. The bibliographical references I provided to improve the quality of the manuscript have not been mentioned or discussed in the manuscript. Therefore, I consider the revised manuscript to be substantially unchanged from the original.
I would like to emphatically draw the attention of the authors to the fact that the review process is aimed at improving the quality of the manuscript. I continue to have the following critical comments:
In the Introduction, the authors assert that "catheter ablation is regarded as first-line therapy for the treatment of symptomatic AVNRT, and slow-pathway ablation or modification is an accepted ablation endpoint." This statement is incorrect (10.1016/j.jacep.2018.09.012). I understand the methodology of the study, but that does not override international standards for standard endpoints for AV nodal slow-path ablations, which are tachycardia non-inducibility or a maximum of 1 echo beat.
The questions "How many operators conducted the ablations, and what was their level of experience in slow pathway ablations utilizing electroanatomical mapping systems? Was intracardiac ultrasound (ICE) employed in any cases?" were answered. However, these details are not addressed in the manuscript, which should be very important due to scientific transparency.
The use of ICE has demonstrated enhanced procedural outcomes in slow pathway ablations compared to a strategy employing fluoroscopy alone (cryo: 10.1186/s12947-019-0162-2; RF: 10.1007/s10840-022-01126-y) and in comparison to electroanatomical mapping systems as well (doi.org/10.3390/jcm12175577). The manuscript should address the role of ICE in slow pathway ablations. All of the recommended literature should be discussed in the Discussion.
The importance of reducing radiation exposure and achieving minimal/zero fluoroscopy procedures should be much more highlighted. The authors should discuss all of the recommended literature in the Discussion. (DOI: 10.1093/eurheartj/ehw223; 10.1111/j.1540-8159.2007.00701.x; 10.1016/j.ijcard.2017.01.128; 10.1007/s00392-018-1220-8; 10.3389/fcvm.2022.856145)"
Reviewer 3 Report
I appreciate authors' efforts in revise the manuscript that now resulted improved. Nevertheless, I would like the Authors could address to mi previous comment #4 (I would appreciate a table describing symptoms before and after ablation in patients with symptoms recurrence at follow-up. Authors could be inspired by table 3 of the following article “Cryoablation of para-Hisian and mid-septal accessory pathways: long-term outcomes of a specific stepwise cryoablation protocol. Minerva Cardiol Angiol. 2023 Jun;71(3):333-341. doi: 10.23736/S2724-5683.22.06136-1”. Moreover, the mentioned article, despite focusing on septal accessory pathway cryoablation, focuses the attention on symptoms improvement as main outcome after ablation. I suggest authors to mention this article to corroborate the message that clinical outcome plays a more relevant role respect to a “pure” electrophysiological point of view, particularly in pediatric/young adult population.) Hoping to help the authors, here attached the manuscript page including Table 3 to the suggested manuscript. Unfortunately, I can not send the whole manuscript for copyright issue. I hope the authors could find if in orders to be inspired for the above mentioned thought
